# MiniDrive: More Efficient Vision-Language Models with Multi-Level 2D Features as Text Tokens for Autonomous Driving

## Abstract

Vision-language models (VLMs) serve as general-purpose end-to-end models in autonomous driving, performing subtasks such as prediction, planning, and perception through question-and-answer interactions. However, most existing methods rely on computationally expensive visual encoders and large language models (LLMs), making them difficult to deploy in real-world scenarios and real-time applications. Meanwhile, most existing VLMs lack the ability to process multiple images, making it difficult to adapt to multi-camera perception in autonomous driving. To address these issues, we propose a novel framework called MiniDrive, which incorporates our proposed **F**eature Engineering Mixture of **E**xperts (FE-MoE) module and **D**ynamic **I**nstruction Adapter (DI-Adapter). The FE-MoE effectively maps 2D features into visual token embeddings before being input into the language model. The DI-Adapter enables the visual token embeddings to dynamically change with the instruction text embeddings, resolving the issue of static visual token embeddings for the same image in previous approaches. The DI-Adapter enables the FE-MoE to further extract and process 2D visual features based on user instructions, focus on attention regions, and reduce redundancy. Compared to previous works, MiniDrive achieves state-of-the-art performance in terms of parameter size, floating point operations, and response efficiency, with the smallest version containing only 83M parameters.

## 1 Introduction

As large-scale pretraining techniques develop, vision-language models (VLMs), due to their powerful visual reasoning capabilities, become the primary choice for visual question answering tasks across various domains. Similarly, in the field of autonomous driving, question-answering reasoning based on VLMs has the potential to become a new method of interaction between drivers and vehicles. This natural language question-answering approach enhances the interpretability of autonomous driving. VLMs integrate perception, prediction, and decision-making during driving into a unified model within autonomous driving systems, functioning as an end-to-end general model for solving various sub-tasks in autonomous driving. Numerous VLMs applications in autonomous driving systems already exist, where these models begin to perform tasks such as closed-loop control, scene perception, and traffic agent behavior analysis in autonomous systems (Chen et al., 2023; Mao et al., 2023; Sima et al., 2023; Xu et al., 2023).

VLMs primarily consist of two main modules, including a vision encoder and an LLM for text generation. This implies that deploying VLMs in a system requires high computational costs and hardware resources. In autonomous driving systems, developing VLMs that consume fewer resources, have lower computational costs, and respond faster becomes a key consideration for practical deployment. However, current research on multimodal large models in autonomous driving mainly focuses on models with over a billion parameters, such as BLIP-2 (Li et al., 2023), LLaMA-7B (Touvron et al., 2023), GPT-3.5, and GPT-4 (Achiam et al., 2023), with the vision encoders relying on pretrained models based on the Transformer architecture, like CLIP (Radford et al., 2021). This consumes substantial computational resources and hardware costs, and requires longer response times, making them challenging to apply and deploy in practice. Recently, EM-VLM4AD(Gopalkrishnan et al., 2024) introduces a lightweight architecture, attempting for the first time to apply lightweight

models in the field of autonomous driving and achieving excellent results. However, there remains a certain gap in response performance compared to billion-parameter models like DriveLM-Agent (Sima et al., 2023). Additionally, autonomous driving typically involves multiple images from different angles, such as front, front-right, front-left, rear, rear-right, and rear-left. Most existing VLMs are trained on single images, making them unsuitable for inputting multiple driving scene images.

To address these challenges, this paper introduces a novel vision-language model called MiniDrive. Unlike traditional mainstream visual-language models, MiniDrive is not a unified model based on the Transformer architecture. We use the efficient backbone network model UniRepLKNet (Ding et al., 2024), which is based on large convolutional kernels, as the vision encoder. We propose the Feature Engineering Mixture of Experts (FE-MoE) and the Dynamic Instruction Adapter (DI-Adapter) to sequentially process visual features and obtain visual tokens before inputting them into the language model. Specifically, UniRepLKNet captures the 2D features of images, and FE-MoE processes multiple 2D features, mapping them into text tokens for input into the language model without requiring stage-wise training for cross-modal fine-grained alignment. Additionally, the DI-Adapter is introduced to enable the mapped visual tokens (i.e., text tokens used as input to the language model) to dynamically adapt to user text instructions, effectively aiding cross-modal understanding between text and images. As shown in Figure 1(a), MiniDrive processes multiple input images along with user instructions to generate natural language responses. It encompasses the most critical capabilities in autonomous driving, including perception, planning, and prediction question-answering abilities. In Figure 1(b), we illustrate that MiniDrive is a lightweight visual-language model with a minimal parameter size, memory footprint, and FLOP count. It can be fully trained with multiple instances on a single RTX 4090 GPU with 24GB of memory. For instance, MiniDrive$_{224}$ has only 83M parameters and a FLOP count of merely 5.9B, which is significantly lower than current visual-language models used in autonomous driving. In terms of response performance, MiniDrive outperforms a series of previous models in question-answering capabilities. Notably, its response quality exceeds that of models with billions of parameters. Additionally, MiniDrive supports both single and multiple image inputs. In Figure 1(c), MiniDrive outperforms open-source models with 7B parameters and above on the single-image evaluation system CODA-LM (Li et al., 2024), approaching the performance of closed-source commercial models. Here are our main contributions:

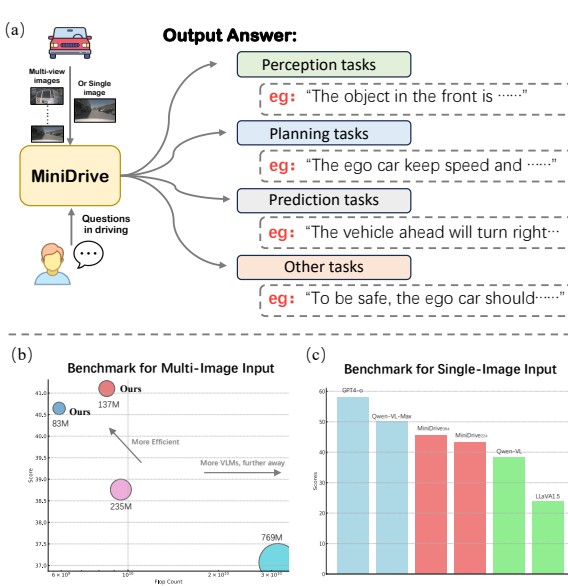

Figure 1: **(a)** shows the input format of MiniDrive and the tasks it can perform. **(b)** compares the average evaluation of multiple-image inputs on the Drive-LM evaluation system with related models. **(c)** compares the average evaluation of single-image inputs on the CODA-LM evaluation system with related models. Minidrive outperforms open-source models larger than 7B and approaches the performance of commercial models.

1. We develop autonomous driving VLMs—MiniDrive, which address the challenges of efficient deployment and real-time response in VLMs for autonomous driving systems while maintaining excellent performance. The training cost of the model is reduced, and multiple MiniDrive models can be fully trained simultaneously on an RTX 4090 GPU with 24GB of memory.

2. MinDrive is attempting for the first time to utilize a large convolutional kernel architecture as the vision encoder backbone for autonomous driving vision-language models, enabling more efficient and faster extraction of 2D features at different image levels. We propose

Feature Engineering Mixture of Experts (FE-MoE), which addresses the challenge of efficiently encoding 2D features from multiple perspectives into text token embeddings, effectively reducing the number of visual feature tokens and minimizing feature redundancy.

3. This paper introduces the Dynamic Instruction Adapter through a residual structure, which addresses the problem of fixed visual tokens for the same image before being input into the language model. The DI-Adapter enables visual features to dynamically adapt to different textual instructions, thereby enhancing cross-modal understanding.

4. We conduct extensive experiments on MiniDrive, achieving state-of-the-art performance compared to autonomous driving VLMs with multi-view image inputs on Drive-LM. Further, we outperform general open-source VLMs (7B) with single-image inputs on CODA-LM by an average of 13.2 points. We open-source all our resources to foster community development.

## 2 RELATED WORK

### 2.1 VISION-LANGUAGE MODELS

The success of the Transformer architecture (Vaswani, 2017) drives the development of LLMs. In the field of computer vision, Dosovitskiy et al. (2020) propose the Vision Transformer (ViT), which divides images into patches and processes them based on the Transformer architecture, adapting it to computer vision tasks with success. Both images and natural language can be effectively learned and represented by the Transformer architecture. A pioneering work is CLIP (Radford et al., 2021), which uses contrastive learning for image-text alignment training, demonstrating superior zero-shot capabilities in image classification tasks. Llava (Liu et al., 2024b) freezes CLIP's vision encoder (ViT) and adds a linear projection layer between the vision encoder and LLMs, aiming to map visual output representations into textual space. Similarly, BLIP-2 (Li et al., 2023) aligns visual and textual representations through a more complex Q-Former. InstructBLIP (Panagopoulou et al., 2023) builds on BLIP-2 with instruction fine-tuning on public visual question-answering datasets. MiniGPT-4 (Zhu et al., 2023) combines a frozen vision encoder and Q-Former with the similarly frozen LLM Vicuna, aligning them with a single projection layer. Llava-1.5v (Liu et al., 2024a) achieves state-of-the-art performance in 11 benchmarks by using CLIP-ViT-L-336px with a multilayer perceptron (MLP) projection layer and adding VQA data tailored for academic tasks with simple response formatting prompts, significantly improving data efficiency. Phi-3-mini (Abdin et al., 2024) features a default 4K context length and introduces a version extended to a 128K context length using LongRope technology, while employing a block structure similar to Llama-2 and the same tokenizer, enabling a lightweight multimodal model. Despite the powerful capabilities of these multimodal large models and their trend toward lightweight design, their parameter counts exceed one billion, making deployment and real-time use on many hardware platforms challenging. Therefore, research and development of efficient vision-language models with smaller parameter sizes and lower computational costs are necessary.

### 2.2 AUTONOMOUS DRIVING BASED ON LLMs

LLMs effectively enhance both the explainability of autonomous driving systems and their interaction with humans (Greer & Trivedi, 2024). These advantages lead researchers to incorporate multimodal data from autonomous driving into LLMs' training, aiming to build multi-modal large models for autonomous driving. Chen et al. (2023) aligned vectorized modal information with LLaMA-7B (Touvron et al., 2023) to train a question-answering model for autonomous driving. The training process follows a two-stage approach: in the first stage, vector representations are aligned with a frozen LLaMA, while in the second stage, LoRA (Hu et al., 2021) is used to fine-tune the language model. DriveGPT4 (Xu et al., 2024) also employs LLaMA as its large language model, using CLIP as the visual encoder. It generates corresponding answers by inputting both visual and textual information. DriveGPT4 leverages ChatGPT/GPT-4 to generate an instruction dataset and trains on this dataset. However, DriveGPT4 only uses single-perspective images, limiting its ability to handle more comprehensive understanding in autonomous driving scenarios. Wang et al. (2023) developed DriveMLM , which uses LLaMA-7B as the foundational language model and ViT-g/14 as the image encoder. This model processes multi-view images, LiDAR point clouds, traffic rules, and user commands to achieve closed-loop driving. Inspired by the chain-of-thought approach in large language

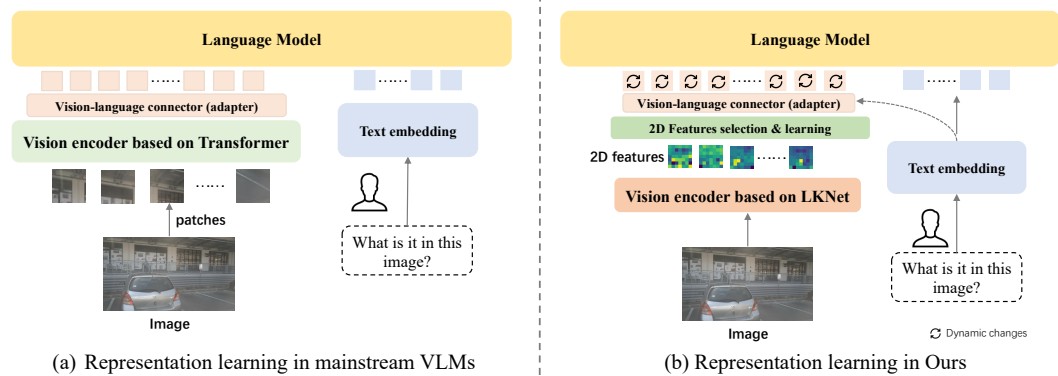

Figure 2: **Comparison of the MiniDrive architecture with mainstream architectures.** (a) Existing vision-language models primarily use a Transformer-based visual encoder to learn image patches as visual tokens. These visual tokens remain unchanged regardless of the user's questions. (b) Our architecture employs a more efficient large convolutional kernel as the visual encoder, learning 2D features of the image as visual tokens. These visual tokens change in response to different user questions.

models (Wei et al., 2022), Sha et al. (2023) proposed a chain-of-thought framework for driving scenarios, using ChatGPT-3.5 to provide interpretable logical reasoning for autonomous driving. Mao et al. (2023) introduced GPT-Driver, which uses ChatGPT-3.5 to create a motion planner for autonomous vehicles, and GPT-Driver reframes motion planning as a language modeling task by representing the planner's inputs and outputs as language tokens. Sima et al. (2023) released the DriveLM dataset, a graphic visual question-answering dataset with question-answer pairs related to perception, behavior, and ego-vehicle planning, based on multi-view image data from the NuScenes dataset (Caesar et al., 2020). To establish baselines, Li et al. (2023) fine-tuned BLIP-2 on this new dataset. EM-VLM4AD (Gopalkrishnan et al., 2024) introduced Gated Pooling Attention (GPA), which aggregates multiple images into a unified embedding and connects it with text embeddings as input to LLMs, achieving promising results on the DriveLM dataset.

While existing work provides significant value and demonstrates strong capabilities for autonomous driving, most models have over a billion parameters. They are largely based on large-scale language models such as GPT-3.5 and LLaMA, and rely on vision encoders built on the ViT architecture, such as CLIP, ViT-g/14, and ViT-B/32. This results in high computational costs, making these models unsuitable for online scenarios. Although there is a trend towards developing lightweight models for autonomous driving, their performance still falls short compared to larger models. In Figure 2, we summarize the architectures of the current mainstream vision-language models and compare them with the architecture of MiniDrive. Existing vision-language models primarily divide images into several patches using a Transformer-based visual encoder, learning each patch as tokens for the input language model. Additionally, during inference, the visual tokens remain fixed regardless of how the user's query changes. Our architecture extracts multi-level 2D features from the image using a vision encoder based on large convolutional kernels (LKNet), further extracting and learning these features to map them as tokens for the input language model. CNNs with large convolutional kernels are more efficient and lightweight (Ding et al., 2022; 2024). Meanwhile, changes in the user's query dynamically alter the visual tokens.

## 3 METHOD

MiniDrive is a vision-language model in the field of autonomous driving, designed to perform visual question answering tasks. It generates text responses by receiving an image and user instruction text as input. In this section, we first provide a detailed introduction to the overall framework of MiniDrive, followed by a specific explanation of the technical details and principles of each module, including the Vision Encoder, **F**eature **E**ngineering Mixture of Experts (FE-MoE) , and **D**ynamic **I**nstruction Adapter (DI-Adapter).

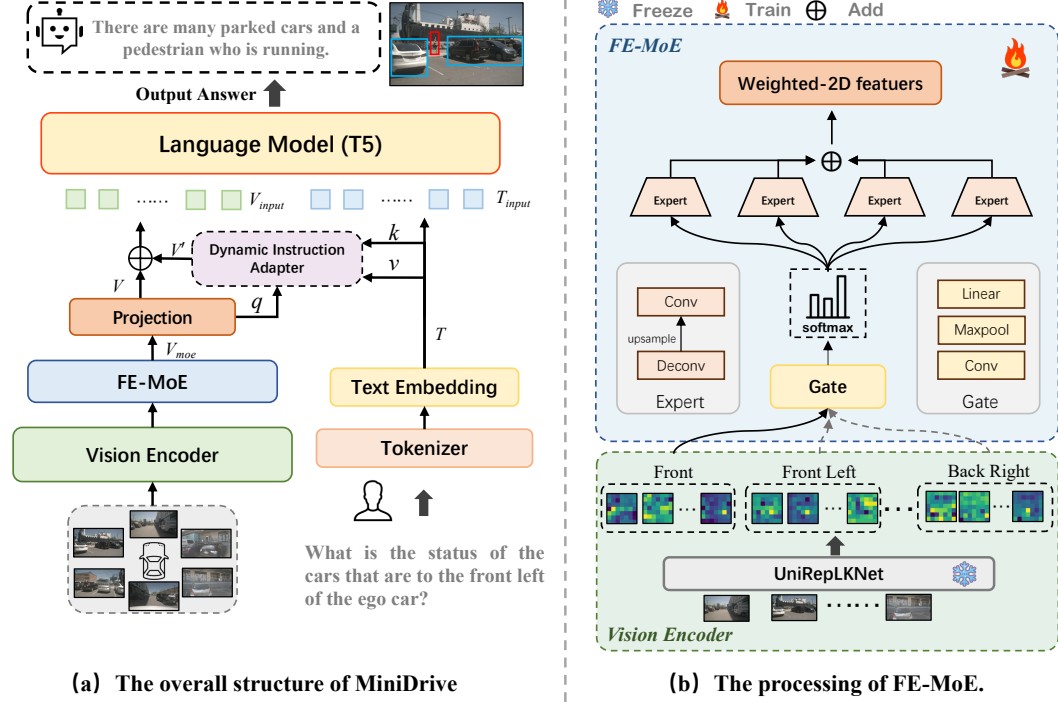

**(a) The overall structure of MiniDrive**   **(b) The processing of FE-MoE.**

Figure 3: **MiniDrive Structural Details.** In Figure (a), the overall architecture of MiniDrive is presented. The image features from the vision encoder input are processed by the FE-MoE and DI-Adapter with residual connections, resulting in visual token embeddings. These embeddings, along with text embeddings, are then fed into the T5-Small language model, producing the output. In Figure (b), the specific framework of FE-MoE is shown. The image is input into UniRepLKNet, producing feature maps at different levels. These feature maps are then fed into the FE-MoE module, where the Gate network generates weights. The 2D visual features are further assigned to different experts for feature mapping and weighted summation.

## 3.1 MODEL ARCHITECTURE

Figure 3 (a) illustrates the overall structure of MiniDrive. In MiniDrive, there are primarily two branches: vision and text. On the vision side, given n images from an autonomous vehicle as input to the visual encoder, $\mathbb{R}^{3 \times H \times W}$, each image receives a set of deep 2D feature representations $V_{2D} \in \mathbb{R}^{c \times h \times w}$. These features are then input into the FE-MoE, where multiple experts compress the information along the channel dimension $c$ and expand it along the height $h$ and width $w$ dimensions to generate new 2D feature representations. In the FE-MoE, the Gate network determines which experts are more suitable for processing each image, assigning different weight values to each expert. Finally, the new 2D feature representations are combined through a weighted sum to produce the new feature set $V_{moe} \in \mathbb{R}^{c' \times h' \times w'}$. Flatten $V_{moe}$ to obtain $V \in \mathbb{R}^{l_1 \times dim_1}$, where the length $l_1$ corresponds to the previous $c'$, and the dimension $dim1$ corresponds to the previous $h' \times w'$. Then, the Projection layer maps $dim_1$ to $dim$, resulting in $V \in \mathbb{R}^{l_1 \times dim}$.

On the text side, the user's natural language instruction is processed through a Tokenizer and Embedding layer to obtain the token embeddings of the text $T \in \mathbb{R}^{l_2 \times dim}$. The embedded sequence of the text $T$ is used as the key (k) and value (v), while the visual embedding sequence $V$ at this stage is used as the query (q). These are fed into the DI Adapter to compute a new visual embedding sequence $V_1$, which now incorporates the contextual information from the text embedding $T$, enabling better cross-modal understanding or decision-making. $V_1$ is then combined with $V$ through a residual connection to form the sequence $V_{input}$, while $T$ is treated as $T_{input}$. The concatenation $[V_{input}, T_{input}]$ is then used as input to the language model. The language model decodes to generate a word sequence with the highest predicted probability. The entire framework efficiently processes multi-image input information, dynamically responding to user queries.

## 3.2 VISION ENCODER

As shown in Figure 3(b), the backbone network of the Vision Encoder is based on the large-kernel neural network UniRepLKNet (Ding et al., 2024), which demonstrates excellent performance across multiple modalities. It effectively leverages the characteristics of large-kernel convolutions, enabling a wide receptive field without the need to go deep into the network layers. While maintaining efficient computation, it also achieves or surpasses the performance of current state-of-the-art techniques across various tasks. This generality and efficiency make it a powerful model with potential in a wide range of perception tasks. A brief review of the overall architecture of UniRepLKNet, as shown in Figure 4, reveals that it primarily consists of multiple sequentially connected Stage layers. Each Stage is mainly composed of a series of Lark Blocks and Smak Blocks. In MiniDrive, we use UniRepLKNet as the backbone of the vision network, where an image is input and the output feature map $F1 \in \mathbb{R}^{c \times h \times w}$ is obtained from the final Stage n.

## 3.3 FEATURE ENGINEERING MIXTURE OF EXPERTS

In Figure 3(b), we present the specific structure of the FE-MoE, which is designed to handle 2D input features from multiple images. Each input image corresponds to a feature map $F_1 \in \mathbb{R}^{c \times h \times w}$ output by the Vision Encoder. To further process the 2D feature representations of each image efficiently, they are input into the FE-MoE. First, $F_1$ is used by the Gate network to obtain the expert selection weights corresponding to the sample. The Gate network mainly consists of convolutional layers, max-pooling layers, and linear layers, as shown in the following equation:

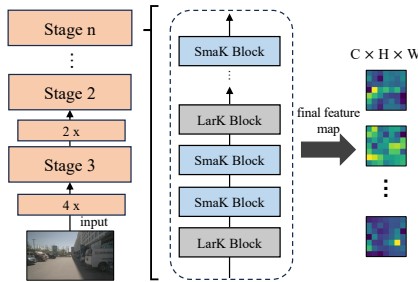

$$Weights = Softmax(Gate(F_1)). \quad (1)$$

Then, $F_1$ passes through each expert network, resulting in a new feature representation $F_2 \in \mathbb{R}^{c' \times h' \times w'}$. Each expert network mainly consists of a deconvolutional layer, a ReLU layer, and a convolutional layer. The deconvolu-

**Figure 4: UniRepLKNet generates feature maps.** We obtain the set of feature maps from each image propagated to the final stage.

tional layer first performs an initial upsampling mapping, increasing the dimensions of the feature map's width and height to expand the amount of information, facilitating subsequent mapping learning. At the same time, it reduces the number of channels in the original feature map to minimize data redundancy and select the most important 2D feature representation information, significantly simplifying the number of subsequent visual tokens. The convolutional layer further transforms the features to enhance the learning capacity of the experts. The formula is shown as follows:

$$F_2 = Conv(ReLu(Deconv(F_1))), \quad (2)$$

$$F_1 \in R^{c \times h \times w} \to F_2 \in R^{c \downarrow \times h \uparrow \times w \uparrow}$$
$$= F_2 \in R^{c' \times h' \times w'}, \quad (3)$$

where, $c \downarrow$ denotes a decrease in the number of channels, while $h \uparrow$ and $w \uparrow$ indicate an increase in the height and width of the feature map, respectively. In this context, $F_2$ represents the output of an individual expert. Given that the weight for the $i$-th expert for an image is $W_i$, and the output from this expert is $F_i$, with the total number of experts being $N$, the feature $V_{moe}$ of the image after processing by the FE-MoE model is expressed by the following formula:

$$F_i = Expert_i(VisionEncoder(Image)), \quad (4)$$

$$V_{moe} = \sum_{i=1}^{N} W_i \cdot F_i. \quad (5)$$

### 3.4 DYNAMIC INSTRUCTION ADAPTER

In previous vision-language models, image representations are fixed before being input into the language model, and they correspond to various text representations before entering the language model for computation. To enable image representations to dynamically transform according to different text representations before being input into the language model, thereby improving cross-modal understanding, we introduce the Dynamic Instruction mechanism and design the Dynamic Instruction Adapter. We use the text input sequence $T$ as the key ($k$) and value ($v$), and the image input sequence $V$ as the query ($q$). Through cross-attention, we compute the fused sequence $V'$ that incorporates textual contextual information. The formula is shown as follows:

$$V' = \text{CrossAtt.}(q = V, k = T, v = T). \tag{6}$$

The sequence in the residual channel is connected via a residual connection with the output sequence of the projection layer, serving as the visual representation prior to the input into the language model. The training of additional language model outputs can be found in the appendix.

## 4 EXPERIMENTS

In this section, we conduct extensive experiments on MiniDrive and analyze the experimental results, including the analysis of quantitative results, computational efficiency, and examples. Finally, ablation experiments are performed to verify the effectiveness of the module.

### 4.1 EXPERIMENTAL SETTINGS

**Datasets** We conduct experiments on the DriveLM dataset. To ensure the fairness of the experiments, we use the same training and evaluation protocol as EM-VLM4AD on the DriveLM dataset, which includes the same training, validation, and test sets. The training set contains approximately 340,184 different multi-view/QA pairs, while the test set and validation set each contain 18,899 different multi-view/QA pairs.

**Models** We construct different versions of MiniDrive based on various UniRepLKNet models as the vision backbone, with the main difference being their ability to learn visual token embeddings. We use UniRepLKNet-A as the vision backbone for processing images with a resolution of 224×224, and UniRepLKNet-S for processing images with a resolution of 384×384. We use the T5-small language model as the foundation.

**Evaluation metrics** To ensure the fairness and reproducibility of the evaluation, we use the same evaluation method as EM-VLM4AD on the DriveLM dataset, assessing the model from four different perspectives: BLEU-4 (Papineni et al., 2002), ROUGE-L (Lin, 2004), METEOR (Banerjee & Lavie, 2005), and CIDEr (Vedantam et al., 2015).

**Implementation details** Each model is trained on a single RTX 4090 GPU. The vision encoder is frozen, while the other parameters are trained with an initial learning rate of 1e-4 and a weight decay of 0.05. Each model is trained for 6 epochs on the training set. Note that in subsequent experiments, MiniDrive refers to the MiniDrive$_{224}$ version by default, with the number of tokens per image set to 16 and the number of experts set to 4.

### 4.2 QUANTITATIVE RESULTS

In Table 1, we compare the evaluation results of MiniDrive with previous works on the test set, including EM-VLM4AD (Gopalkrishnan et al., 2024) and Drive-Agent (Sima et al., 2023). In terms of overall performance on the metrics, both MiniDrive$_{224}$ and MiniDrive$_{384}$ outperform previous methods, although DriveLM-Agent surpasses us in BLEU-4, its parameter count is significantly larger than ours, reaching 3.96B.

### 4.3 COMPUTATIONAL ANALYSIS

In this section, we primarily compare the differences between MiniDrive and a range of existing vision-language models in terms of parameter count, Floating Point Operations (FLOPs), and memory usage (GB). The results are shown in Table 2. Using an input image resolution of 224 as an example, MiniDrive demonstrates superior performance in all three aspects.

Table 1: Performance on DriveLM. We compare the response performance of different models on the same test set. **Bold** indicates the highest value, while an underline indicates the second-highest value.

| Method | Ref. | DriveLM | | | |
|---|---|---|---|---|---|
| | | **BLEU-4** ↑ | **METEOR** ↑ | **ROUGE-L** ↑ | **CIDEr** ↑ |
| EM-VLM4AD$_{Base}$ | CVPR' 24 | 45.36 | 34.49 | 71.98 | 3.20 |
| EM-VLM4AD$_{QLarge}$ | CVPR' 24 | 40.11 | 34.34 | 70.72 | 3.10 |
| DriveLM-Agent | ECCV' 24 | **53.09** | 36.19 | 66.79 | 2.79 |
| MiniDrive$_{224}$ (Ours) | - | 49.70 | 36.30 | 73.30 | 3.28 |
| MiniDrive$_{384}$ (Ours) | - | 50.20 | **37.40** | **73.50** | **3.32** |

Table 2: The computational analysis of the model includes a comparison of the parameter size, floating point operations (FLOPs), and GPU memory usage.

| Model | Parameters | FLOPs | Memory (GB) |
|---|---|---|---|
| DriveMLM | 8.37B | 535B | 36 |
| Drive-GPT4 | 7.3B | 329B | 29.2 |
| LLM-Driver | 7B | 268B | 28 |
| DriveLM-Agent | 3.96B | 439B | 14.43 |
| EM-VLM4AD$_{Base}$ | 345M | 9.9B | 1.97 |
| MiniDrive$_{224}$ (ours) | 83M | 5.9B | 1.03 |

## 4.4 QUALITATIVE EXAMPLES

In Figure 5, we present the actual responses of MiniDrive on unseen samples across three different tasks. To provide an interpretability analysis of MiniDrive's perception of multi-view image inputs, we analyze the activation maps of MiniDrive in various scenarios. In Figure 5 (a), MiniDrive demonstrates perceptual question-answering for multiple image inputs, with the blue box indicating the image referenced by the user's instruction for the position "back left." The red box corresponds to MiniDrive's response, primarily focusing on that image, identifying "many cars, one truck, and one pedestrian" at the specified location. In Figure 5 (b), MiniDrive demonstrates planning question-answering for multiple image inputs. Based on the user's instruction and the spatial term "CAM_FRONT", MiniDrive focuses on the red box on the left side of the corresponding front image. This attention aligns with the elements that humans consider when making planning decisions, including the traffic lane markings and vehicles on the left side of the ego car. In Figure 5 (c), MiniDrive demonstrates predictive question-answering for multiple image inputs. Based on the user's instruction to predict the movement of the pedestrian in the "front left" position, MiniDrive focuses on the pedestrian in the corresponding positional image, highlighted by the red box. Taken together, the objects that MiniDrive focuses on in the activation map align with the reasoning followed by human drivers during driving, indicating that MiniDrive possesses a certain level of reliability and interpretability.

## 4.5 ABLATION STUDIES

To validate the effectiveness of each module, we design a series of ablation experiments. In Table 3, we investigate the impact of FE-MoE and Dynamic Instruction Adapter (DI-Adapter) on MiniDrive. When FE-MoE and Dynamic Instruction Adapter are introduced separately, the results of various metrics improve, and when both modules are introduced simultaneously, a better effect is achieved. This indicates the effectiveness of the mechanisms between the modules. The details of other ablation experiments can be found in the appendix.

## 5 FURTHER ANALYSIS

Although MiniDrive is designed as an autonomous driving question-answering model for receiving multi-image inputs, it extracts, compresses, and re-learns the information from multiple images as Text Tokens for the language model. However, it can still be used for single-image input tasks. We compare it with existing mainstream open-source and closed-source general models on CODA-LM, as shown in Table 4. It is evident that despite MiniDrive having only 83M parameters, it demon-

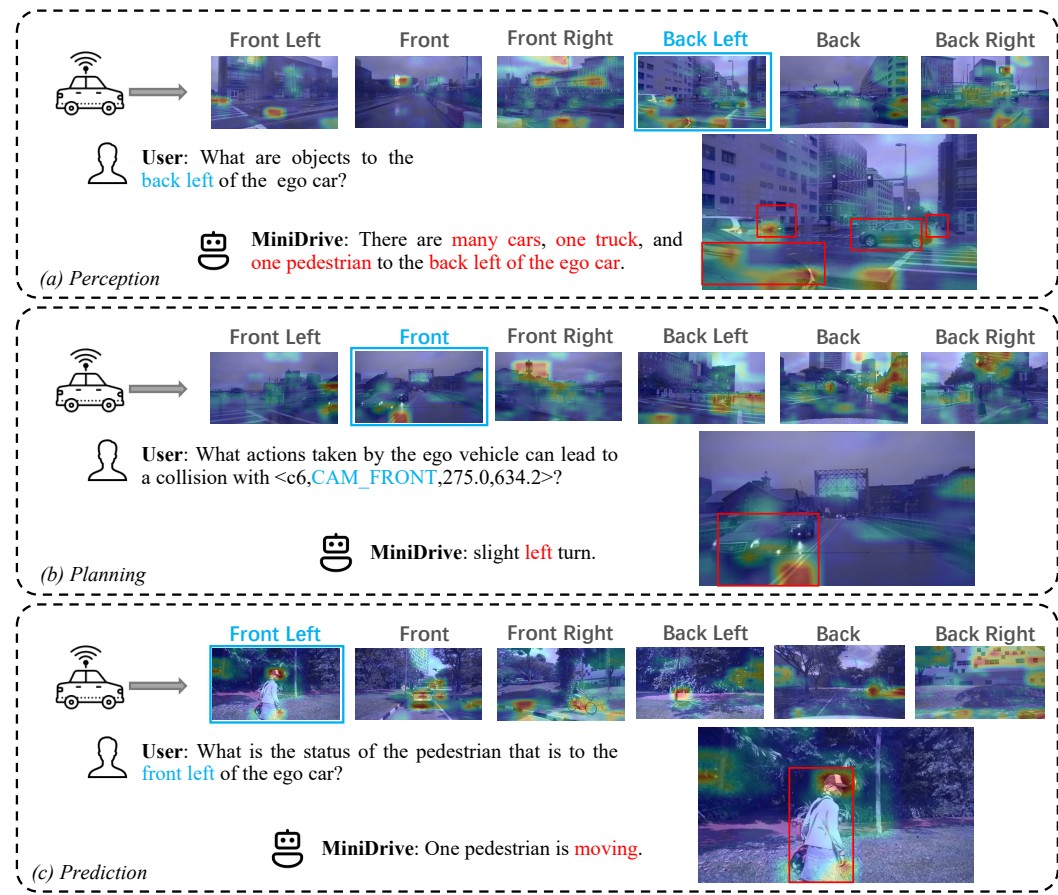

Figure 5: Examples of MiniDrive's Response. The color blue represents the user command querying for multi-image input. The color red represents the activation response generated by MiniDrive corresponding to the text.

Table 3: Ablation among modules. We compare the response performance of different models on the same test set. **Bold** indicates the highest value, while an underline indicates the second-highest value.

| FE-MoE | DI-Adapter | DriveLM | | | |
|---|---|---|---|---|---|
| | | **BLEU-4 ↑** | **METEOR ↑** | **ROUGE-L ↑** | **CIDEr ↑** |
| – | – | 45.70 | 34.09 | 69.74 | 3.07 |
| ✓ | – | 48.30 | 35.40 | 72.10 | 3.23 |
| – | ✓ | 48.00 | 35.70 | 72.00 | 3.16 |
| ✓ | ✓ | **49.70** | **36.30** | **73.30** | **3.28** |

strates superior performance, outperforming open-source models and approaching the performance of closed-source models. Due to the issue with the distribution of the training data, we believe that this is the main factor contributing to MiniDrive's strong ability to recognize "Cone". Further details can be found in the appendix.

## 6    CONCLUSION

In this paper, we present MiniDrive, a state-of-the-art lightweight vision-language model for autonomous driving. We introduce the FE-MoE and DI-Adapter mechanisms, proposing a novel approach that maps 2D convolutional features into text tokens for language models. Our model achieves outstanding results on two datasets, DriveLM and CODA-LM. In the future, we aim to

Table 4: Performance on CODA-LM. MiniDrive is compared with Multimodal Large Language Models. Bold indicates the highest value, while an underline indicates the second-highest value.

| Method | Parameters | General↑ Text-Score | Regional Perception ↑ | | | | | | Suggestion↑ Text-Score |
|---|---|---|---|---|---|---|---|---|---|
| | | | ALL | Vehicle | VRU | Cone | Barrier | Other | |
| LLaVA1.5 (Liu et al., 2024a) | 7B | 22.60 | 34.78 | 40.00 | 28.00 | 32.22 | 24.00 | 10.00 | 14.20 |
| Qwen-VL-Chat bai2023qwen | 7B | 26.00 | 53.33 | 57.76 | 60.00 | 48.89 | 44.29 | 35.71 | 35.40 |
| Qwen-VL-Max (Bai et al., 2023) | api-model | 34.60 | 68.17 | 69.83 | 56.00 | **80.00** | 59.29 | 65.71 | 47.40 |
| GPT-4o (OpenAI, 2024) | api-model | **45.00** | **73.76** | **75.69** | **66.00** | 75.56 | **69.29** | **70.00** | **55.50** |
| MiniDrive$_{224}$(Ours) | 83M | 21.60 | 62.15 | 62.93 | 36.00 | **86.67** | 59.29 | 48.57 | 45.40 |
| MiniDrive$_{384}$(Ours) | 137M | 24.60 | 66.34 | 67.41 | 36.00 | 84.44 | 62.86 | 62.85 | 45.44 |

develop a real-time response model with video input to further advance autonomous driving technology.

## 7 LIMITATIONS

MiniDrive builds VLMs specific to the autonomous driving domain and has achieved excellent results on current mainstream benchmarks. However, it still lacks a certain level of generalization, which we believe is due to the limitations of the training samples. The existing autonomous driving field requires more public datasets and efforts to develop them. Additionally, MiniDrive's training is primarily focused on instruction-based datasets, and it continues to experience hallucination issues.

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

## A  TRAINING

Due to the effectiveness of each module in MiniDrive and the consumption of only a small amount of computational resources, the training employs a straightforward full-parameter approach, meaning all parameters in MiniDrive are included in the training process. Meanwhile, MiniDrive is freezed the vision encoder. MiniDrive is supervised by label text, with loss calculated using cross-entropy, which quantifies the difference between the text sequence generated by the model and the target text. The formula is as follows:

$$\text{Loss} = -\sum_{i=1}^{n} y_i \log(p_i), \quad (3)$$

where $n$ is the number of tokens, $y_i$ is the true label for token $i$, and $p_i$ is the predicted probability for token $i$.

## B  MORE ABLATION STUDIES

We configure the number of tokens per image in MiniDrive$_{224}$ to 16 and set the number of experts to 2, 4, and 6 for testing on DriveLM. Additionally, we configure the number of experts in MiniDrive$_{224}$ to 4 and set the number of tokens per image to 8, 16, and 32 for testing on DriveLM. The results are shown in Figure 6. When the tokens become larger, the language model's ability to learn longer sequences decreases. As the number of experts increases, the training difficulty of the FE-MoE network grows, leading to a decline in learning performance.

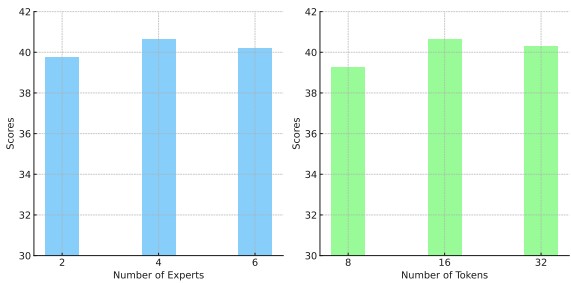

Figure 6: More Ablation Studies on Tokens and Experts.

## C    MORE EXAMPLES

In this section, we demonstrate more response instances of MiniDrive. In Figure 7, we showcase question-answering instances on DriveLM. While in Figure 8, we present question-answering instances on CODA-LM. We train on the official training set provided by CODA-LM and conduct testing on the Mini set. The parameter settings are consistent with those described in the experimental section.

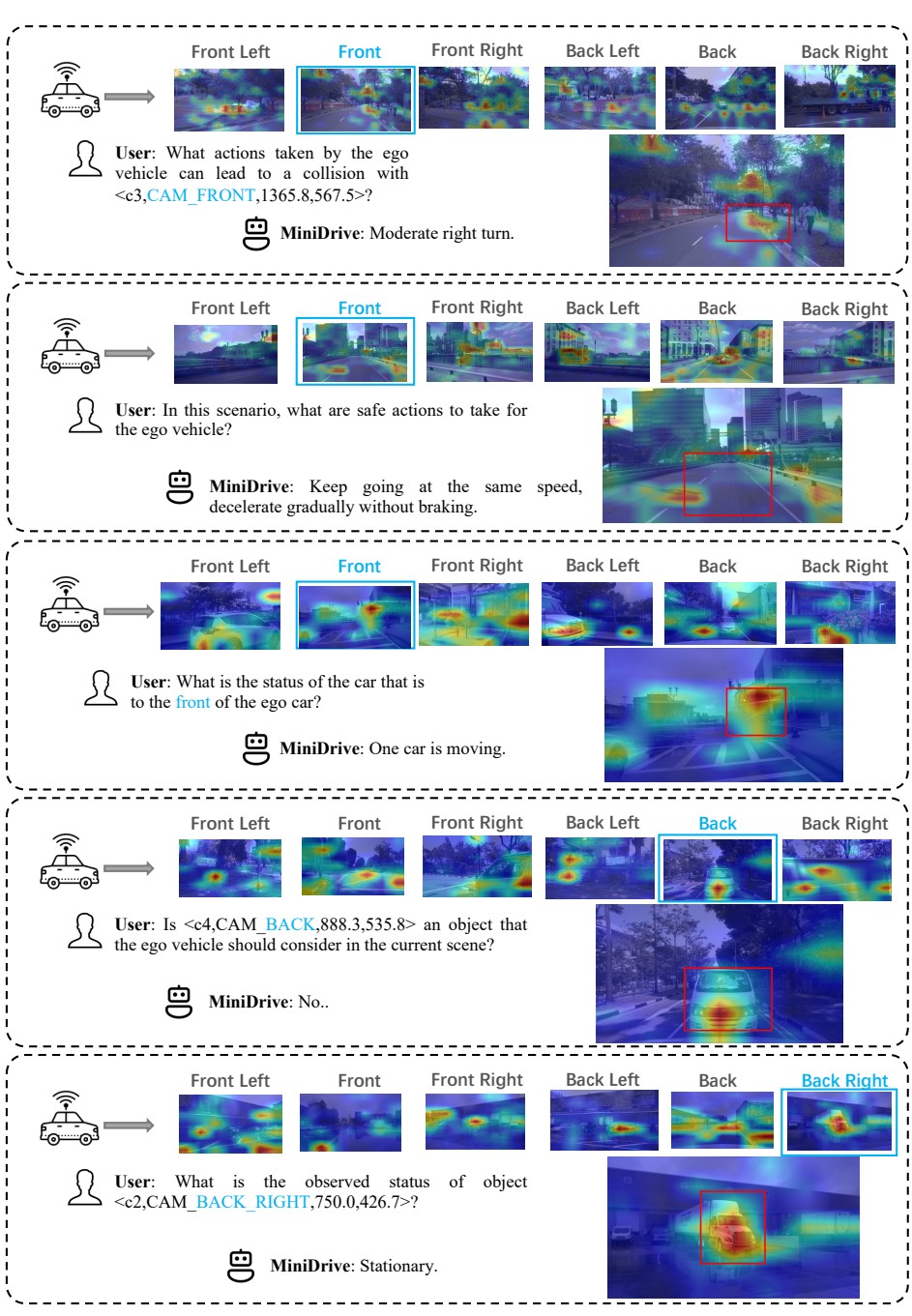

Figure 7: **Examples of MiniDrive's Response on DriveLM.**

756
757
758
759
760
761
762
763
764
765
766

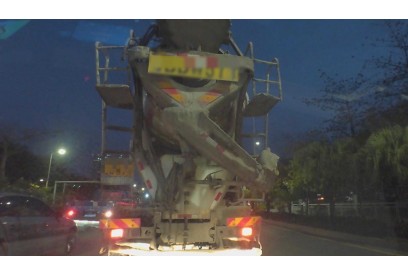

**User:**

There is an image of traffic captured from the perspective of the ego car. Focus on objects influencing the ego car's driving behavior: vehicles (cars, trucks, buses, etc.), vulnerable road users (pedestrians, cyclists, motorcyclists), traffic signs (no parking, warning, directional, etc.), traffic lights (red, green, yellow), traffic cones, barriers, miscellaneous(debris, dustbin, animals, etc.). You must not discuss any objects beyond the seven categories above. Please describe each object's appearance, position, direction, and explain why it affects the ego car's behavior.

**MiniDrive:**

In the traffic image, there is a large truck directly in front of the ego car, occupying the same lane. The truck is characterized by its rear lights and reflective markings. The presence of this truck is significant as it occupies the entire lane and may obstruct the view ahead for the ego car, necessitating a lane change or a lane change to maintain a safe following distance. Additionally, there is a traffic light visible in the distance, which is currently green. This indicates that the ego car has the right of way to proceed. However, the driver should remain alert for any changes in the traffic light status. There are no vulnerable road users, traffic signs, traffic cones, barriers, or other objects present in the image that affect the driving behavior of the ego car.

**Evaluator (gpt-4o-2024-05-13 ):** The prediction text accurately captures the key elements described in the reference text. It correctly identifies the presence of a large truck in front of the ego car and emphasizes the impact this truck has on visibility and the need for a safe following distance. The description of the truck's safety lights and reflective markings aligns well with the information provided in the reference text. The acknowledgment of the green traffic light ahead also matches the reference, along with the requirement for the ego car to remain alert for any changes. Crucially, the prediction text does not include any hallucinated objects or details that were not mentioned in the reference text. Thus, it maintains accuracy and relevance to the driving context described. Overall, the correlation between the objects mentioned and their impacts on driving behavior is consistent between both texts. Therefore, I would rate this response as follows: Rating: [[10]].

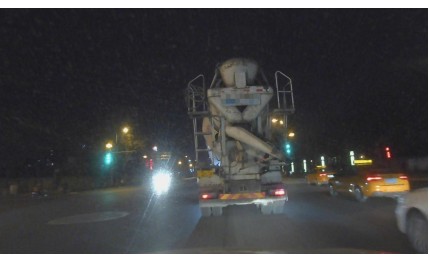

**User:**

There is an image of traffic captured from the perspective of the ego car. Focus on objects influencing the ego car's driving behavior: vehicles (cars, trucks, buses, etc.), vulnerable road users (pedestrians, cyclists, motorcyclists), traffic signs (no parking, warning, directional, etc.), traffic lights (red, green, yellow), traffic cones, barriers, miscellaneous(debris, dustbin, animals, etc.). You must not discuss any objects beyond the seven categories above. Please provide driving suggestions for the ego car based on the current scene.

**MiniDrive:**

The ego car should maintain a safe following distance from the cement mixer truck, be prepared to slow down or stop if the truck's brake lights indicate it is slowing down or stopping, and monitor the white sedan on the left for any lane changes. Additionally, the ego car should be cautious of the motorcyclist on the right and be ready to adjust its position if necessary. The ego car should also be ready to react to the traffic light ahead and be ready to stop if it changes.

**Evaluator (gpt-4o-2024-05-13 ):** The predicted text correctly emphasizes the importance of maintaining a safe following distance from the cement mixer truck and being prepared to stop or slow down. It also includes relevant observations about the white sedan and the motorcyclist. However, it fails to mention the red traffic light explicitly, which is a critical element of the given scenario indicating an imminent stop may be required. While the details about monitoring traffic are valuable, they slightly diverge from the core suggestions of the reference text. Overall, the response is actionable and rational, yet it lacks completeness concerning the traffic light condition. Rating: [[7]]

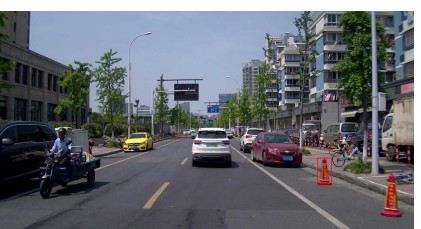

**User:**

Please describe the object inside the red rectangle in the image and explain why it affect ego car driving.

**MiniDrive:**

This object is a traffic cone, typically used to signal temporary changes in traffic patterns or to guide traffic away from certain areas. Its bright color and reflective stripes make it highly visible to drivers. In this scenario, the presence of the traffic cone suggests that there may be road work ahead or a temporary obstruction. The ego car should reduce speed and proceed with caution, possibly preparing to change lanes if the cone is indicating a closed lane or a hazard.

**Evaluator (gpt-4o-2024-05-13 ):** The predicted text accurately describes the function and visibility of a traffic cone, closely mirroring the reference text's content. Both texts explain that traffic cones are used to signal road hazards and guide traffic, emphasizing the safety considerations for vehicles approaching the cones. The language is similar, and the overall message about caution and potential lane changes is consistent across both texts. Rating: [[9]]

Figure 8: **Examples of MiniDrive's Response on CODA-LM.**