# OpenReview forum: "MiniDrive: More Efficient Vision-Language Models with Multi-Level 2D Features as Text Tokens for Autonomous Driving"
_ICLR.cc/2025/Conference — ICLR 2025 Conference Withdrawn Submission_

### Official Review · Reviewer_aSWC · 2024-10-30

**Soundness:** 2
**Presentation:** 2
**Contribution:** 2
**Rating:** 5
**Confidence:** 4

**Summary:**

This paper introduces a lightweight and efficient vision-language model for addressing the QA-based autonomous driving task (DriveLM). Specifically, a framework called MiniDrive is proposed, which incorporates a Feature Engineering Mixture of Experts module (FE-MoE) and a Dynamic Instruction Adaptor (DI-Adapter). The FE-MoE applies an efficient visual encoder, UniRepLKNet, built based on large convolution kernels, for efficient encoding of multi-view images, followed by a mixture of experts similar to multi-head convolution to enhance model expressiveness. The DI-Adapter conditions visual features based on the input instruction. The proposed method is tested on DriveLM and CODA-LM datasets to verify the model performance.

**Strengths:**

1.	The research and idea proposed in this paper are very well-motivated, focusing on the efficiency of the VLM-based approach that is critical to deploying the method in practice. The methods and findings in this paper could inspire relevant research in this direction.
2.	The paper applies the convolutional UniRepLKNet as the visual encoder, instead of ViT-based encoders, to efficiently encode image inputs from multiple directions and introduces an MoE to enhance the representations. The ablation study justifies the effectiveness of this model.
3.	The proposed MiniDrive outperforms other recent works on the DriveLM dataset by a large margin despite having significantly fewer parameters and a simpler training process, which demonstrates the potential of this method.

**Weaknesses:**

1.	The paper argues that the DI-Adapter is one of its main contributions; although it seems effective, as shown in Table 3, the novelty is very limited. The idea of adapting visual representations conditioned on instruction has been extensively considered in VLM literature, e.g., the classic InstructBLIP.
2.	The proposed method encodes observations from six directions for DriveLM QA. However, it is unclear to me how the model utilizes and benefits from cross-image information.
   - From Section 3.3, each image is modeled independently, and all features are fed to the T5 language model together.
   - From the examples provided in this paper, it seems that all questions (except Figure 7 case 2) clearly specify exactly one view for QA. Hence, it is unclear to me if encoding all views is necessary and how much improvement MiniDrive can gain from this.
3.	The biggest weakness of this paper to me is the superficial experiments and lack of in-depth analysis of the system and its behavior.
   - The paper argues that the method has significant efficiency advantages and real-time inference, but there is no experiment to compare the speed with previous approaches. E.g., how many frames/questions can the model process per second (FPS)?
   - It is unclear how exactly the proposed method performs in different tasks, e.g., perception, prediction, and planning, as categorized by the DriveLM dataset, and what is the influence of each proposed component in these tasks.
   - The paper and title highlight the use of multi-level 2D features (the UniRepLKNet), but there is no experiment to compare this encoder to encoders applied in previous works.
   - The DriveLM dataset consists of three components for evaluating Motion, Behavior, and P1-3. This paper only studies P1-3, which includes relatively simple one-shot QA instead of Behavior or Motion, where the traffic anticipation and actual rollout of the ego vehicle are needed. I am not convinced that only evaluate on P1-3 can reflect the actual potential of the proposed method in autonomous driving.
   - A relevant question to the above is how to translate the QA responses to actual vehicle control. It is unclear to me how many questions need to be asked and how to integrate those answers before the agent can make a correct control decision in a complex scenario.

**Questions:**

Most of my questions are included in the Weakness above. I hope the authors can respond concisely to them.

Suggestions: I hope the paper can clarify more about the dataset and task setup.

---

### Official Review · Reviewer_YVQH · 2024-10-31

**Soundness:** 2
**Presentation:** 2
**Contribution:** 2
**Rating:** 3
**Confidence:** 5

**Summary:**

This paper proposes MiniDrive, a vision-language model optimized for autonomous driving tasks. MiniDrive incorporates the Feature Engineering Mixture of Experts (FE-MoE) and Dynamic Instruction Adapter to dynamically handle visual features and integrate contextual information from text inputs. The paper highlights the model’s lower FLOPs, parameter count, and memory usage compared to larger models, aiming for efficient processing on single GPUs

**Strengths:**

1.	Goal-Oriented Design for Autonomous Driving: MiniDrive seeks to address the high resource demands of typical vision-language models, specifically aiming to enable real-time processing in the context of autonomous driving.
2.	Efficient Model Parameters and FLOPs: The model claims efficiency in terms of FLOPs and memory usage, potentially supporting multi-instance training on a single GPU, which can be beneficial for applications with limited computational resources.

**Weaknesses:**

1.	Lack of Real-Time Performance Evaluation: Despite the model’s claim of real-time suitability, there is no specific evaluation of inference time or processing speed, which is critical for applications in autonomous driving. The model’s practical performance remains unproven in real-world settings.
2.	Limited Novelty of FE-MoE: 1. The FE-MoE mechanism in MiniDrive employs a continuous weighted-sum approach across multiple experts, similar to the foundational work on Mixture of Experts by Shazeer et al. (2017). In this pioneering study, Shazeer et al. introduced a sparsely-gated MoE structure where multiple experts contribute via a weighted-sum aggregation. While the original model aimed at efficiency by leveraging sparse gating, MiniDrive’s FE-MoE does not implement sparse gating, thus potentially requiring higher computational resources. Given the similarity in the underlying weighted-sum aggregation concept and the absence of a sparse mechanism, the FE-MoE lacks sufficient differentiation from established MoE architectures and does not clearly demonstrate a unique advantage for autonomous driving. Further clarification on how FE-MoE improves upon traditional MoEs would strengthen the claim of novelty.
3.	Insufficient Multi-Camera Environment Evaluation: While the model mentions multi-image processing, there are no evaluations or specific methodologies provided to demonstrate effectiveness in multi-camera setups, which are essential in autonomous driving for comprehensive scene understanding.
4.	Inadequate Control Experiment Details: In Table 3, the comparison between MiniDrive and a baseline without FE-MoE is presented. However, there is insufficient information on the exact parameter count and FLOPs of the baseline model, raising concerns about the fairness and interpretability of these results.
5.	Minor Errors and Inconsistencies: The paper contains typographical errors (e.g., “ReLu” instead of “ReLU” and inconsistent citation formatting). These, along with unclear baseline setup explanations, detract from the paper’s overall clarity and polish.

**Questions:**

Given the paper’s lack of novelty in its proposed FE-MoE mechanism, absence of real-time performance benchmarks, and incomplete evaluation for autonomous driving environments (e.g., multi-camera setup), its contributions are limited in scope and significance. The paper would benefit from a clearer demonstration of MiniDrive’s practical impact, including concrete real-time performance metrics, and a more thorough investigation of its architectural claims.

1.	Real-Time Performance Validation: Could you provide detailed inference times or benchmarks to clarify MiniDrive’s performance in real-time environments? This would help substantiate the paper’s emphasis on efficiency.
2.	Differentiation of FE-MoE from Existing MoE Architectures: How does FE-MoE specifically improve upon standard MoE frameworks, particularly sparse MoEs, in terms of efficiency for autonomous driving applications?
3.	Multi-Camera Integration Methodology: Could you elaborate on how MiniDrive supports multi-camera setups in autonomous driving, and whether it maintains robust performance across diverse camera angles and resolutions?
4.	Baseline Model Configuration: In Table 3, could you clarify the configuration of the baseline model without FE-MoE, specifically addressing the parameter count and computational requirements to ensure fair comparison?

---

### Official Review · Reviewer_G9wT · 2024-11-01

**Soundness:** 2
**Presentation:** 2
**Contribution:** 2
**Rating:** 5
**Confidence:** 3

**Summary:**

The authors proposed MiniDrive framework, a lightweight vision-language model for autonomous driving, optimizing efficiency with reduced parameters. Using the FE-MoE module for visual processing and the DI-Adapter for dynamic instruction response, it achieves competitive performance on the DriveLM dataset while lowering computational costs, making it practical for real-time use on limited hardware​.

**Strengths:**

1. **Efficiency**: *MiniDrive* is a lightweight VLM with low FLOPs, suitable for real-time deployment on limited hardware, making it highly practical for autonomous driving.

2. **Dynamic Adaptation**: The *Dynamic Instruction Adapter* enhances cross-modal understanding by adapting visual tokens to user instructions, improving interaction quality in real-world applications.

**Weaknesses:**

This paper, though notable, leans more toward an engineering approach than a research-oriented contribution. I identify the following limitations:

1. **Insignificant Training Cost Reduction**: Reducing training cost is not significant. A comparable 4-bit or 8-bit quantized large language model (LLM) with ~7B parameters can also be fine-tuned on a single RTX 4090 GPU using adapters, which limits the novelty in terms of efficiency.

2. **Limited Benchmarking Scope**: The integration of UniRepLKNet for visual feature extraction and the Mixture of Experts (MoE) design should be evaluated on a broader range of benchmarks beyond autonomous driving (AD) datasets. If the authors focus solely on AD datasets, it would be beneficial to emphasize how the architecture uniquely benefits AD scenarios. Currently, the proposed FE-MoE framework appears generalizable to various visual modality applications, lacking a clear advantage for AD-specific use cases.

3. **Lack of Task-Specific Uniqueness in Dynamic Instruction Adapter**: The Dynamic Instruction Adapter is a promising concept, though it suffers from a similar limitation as (Limination 2) — it lacks specialization for AD tasks, which could limit its applicability in scenarios beyond general-purpose visual adaptation. Also, this idea is not new and similar idea is applied in many other works (e.g. Llama-Adapter [1] and CogVLM [2]).

4. **Ambiguity in the MoE Approach**: The FE-MoE’s primary goal seems to be fusing tokens from different camera sources, yet the reasoning behind using a Mixture of Experts is unclear. In most AD scenarios, information from all cameras is essential. Applying a hard limit (e.g., selecting only the top-k experts, where \( k < 6 \)) risks discarding critical visual data from unselected cameras. Conversely, if \( k = 6 \) (i.e., using all cameras), simpler feature transformation and merging techniques could be more efficient than the current gating + softmax + elementwise weighted merge approach, which substantially increases GPU memory consumption.

5. **Simplistic Illustrative Examples**: Figure 5 does not adequately demonstrate the benefits of MiniDrive over competing frameworks, such as DriveLM-Agent. The examples lack complexity and do not showcase significant advantages.

6. **Incomplete Comparative Evaluation**: In Table 2, models like LLM-Driver and Drive-GPT4 possess explicit waypoint prediction capabilities and are thus evaluated with UniAD metrics. MiniDrive, however, seems like has not implemented waypoint prediction, preventing a direct comparison with these models and leaving its performance on this critical aspect unaddressed.

[1] Zhang, Renrui, et al. "Llama-adapter: Efficient fine-tuning of language models with zero-init attention." arXiv preprint arXiv:2303.16199 (2023).

[2] Wang, Weihan, et al. "Cogvlm: Visual expert for pretrained language models." arXiv preprint arXiv:2311.03079 (2023).

**Questions:**

Same as limitations

---

### Note · Authors · 2024-11-13

I have read and agree with the venue's withdrawal policy on behalf of myself and my co-authors.